# Glycosylated IgG antibodies contribute to the recovery of haemorrhagic fever with renal syndrome patients

Chuansong Quan[1,2,3†], Lu Wang[3,4†], Jiming Gao[5†], Yaoni Li[6†], Xiaoyu Xu[7†], Houqiang Li[3], Zixuan Gao[7], Wenxu Ruan[7], Hongzhi Liu[3], Qian Li[3], Weijia Xing[3], Liqiong Zhao[6], Michael J Carr[8,9], Weifeng Shi[10,11]*, Haifeng Hou[3]*

[1]Key Laboratory of Emerging Infectious Diseases in Universities of Shandong, Shandong First Medical University and Shandong Academy of Medical Sciences, Ji'nan, China; [2]The Second Affiliated Hospital of Shandong First Medical University, Taian, China; [3]School of Public Health, Shandong First Medical University and Shandong Academy of Medical Sciences, Ji'nan, China; [4]Medical Records and Statistics Management Office, Tengzhou Central People's Hospital, Zaozhuang, China; [5]Department of Neurobiology and Physiology, School of Clinical and Basic Medicine, Shandong First Medical University and Shandong Academy of Medical Sciences, Ji'nan, China; [6]Baoji Central Hospital, Baoji, China; [7]School of Life Sciences, Shandong First Medical University and Shandong Academy of Medical Sciences, Taian, China; [8]National Virus Reference Laboratory, School of Medicine, University College Dublin, Dublin, Ireland; [9]International Collaboration Unit, International Institute for Zoonosis Control, Hokkaido University, Sapporo, Japan; [10]Ruijin Hospital, Shanghai Jiao Tong University School of Medicine, Shanghai, China; [11]Shanghai Institute of Virology, Shanghai Jiao Tong University School of Medicine, Shanghai, China

*For correspondence:
shiwf@ioz.ac.cn (WS);
hfhou@163.com (HH)

†These authors contributed equally to this work

## eLife Assessment

The authors investigated the potential role of IgG N-glycosylation in Haemorrhagic Fever with Renal Syndrome (HFRS), which may offer significant insights for understanding molecular mechanisms and for the development of therapeutic strategies for this infectious disease. The findings are **valuable** to the field and the strength of evidence to support the findings is **solid**.

**Abstract** Haemorrhagic fever with renal syndrome (HFRS) is a fatal disease caused by Hantaan virus (HTNV) infection. Humoral immunity is essential for effective viral clearance; however, the glycosylation characteristics of immunoglobulin G (IgG) in HFRS patients are not well known. Peripheral blood mononuclear cells from HFRS patients were obtained for B subset analysis using scRNA-seq and flow cytometry. HTNV-specific IgG antibody titers were detected by enzyme-linked immunosorbent assay, and IgG glycosylation was analyzed by ultra-performance liquid chromatography. The proportions of the antibody-secreting memory (ASM) B cells and plasmablasts (PB) were significantly expanded among acute HFRS patients. We discovered significantly increased fucosylated IgG and decreased bisecting N-acetylglucosamine during the convalescent phase of HTNV infection. Meanwhile, positive correlations were observed between ASM subsets and galactosylation/sialylation in the IgG Fc region, and between PB subsets and sialylation. Notably, the glycosylation-related genes, such as *RPN1 and RPN2*, were primarily expressed differentially in

the ASM and PB subclusters, which were enriched in the N-glycosylation modifications of proteins through asparagine. Our findings indicated that IgG N-glycosylation may play a crucial role in combating HTNV infection and contributing to clinical recovery, which provided new insights for optimizing glycoengineered therapeutic antibodies.

## Introduction

The causative agent of haemorrhagic fever with renal syndrome (HFRS) is Hantaan virus, which belongs to the family *Hantaviridae* and is primarily transmitted by rodents (*Zheng et al., 2019*). China is the most heavily affected country by HFRS, accounting for >90% of global cases, and a total of 32,462 cases of HFRS were reported in China during 2019–2022, with a fatality rate ranging from 0.1 to 15% (*Li et al., 2023*) (https://www.chinacdc.cn).

Hantaviruses are tri-segmented, single-stranded, negative-sense RNA viruses, whose genomes consist of three regions: large (L), medium (M), and small (S). The glycoproteins Gn and Gc, encoded by the M segment, can infect target cells - primarily vascular endothelial cells - via β3 integrin receptors (*Pizarro et al., 2019*). Simultaneously, they could also infect other cell types, such as mononuclear macrophages and dendritic cells, leading to systemic viral infection. Although hantavirus replication is thought to occur primarily in the vascular endothelium without direct cytopathic effects, a plethora of innate immune cells mediate host antiviral defenses. These include natural killer cells, neutrophils, monocytes, and macrophages, together with pattern recognition receptors (PRRs), interferons (IFNs), antiviral proteins, and complement activation, e.g., via the pentraxin 3 (PTX3) pathway, which can exacerbate HFRS disease progression leading to immunopathological damage through cytokine/chemokine production, cytoskeletal rearrangements in endothelial cells, ultimately amplifying vascular dysfunction (*Tariq and Kim, 2022*). Rapid and effective humoral immune responses, however, such as neutralizing antibody responses targeting the glycoproteins Gn/Gc, contribute to rapid recovery from HFRS and are critical for protection from severe disease (*Engdahl and Crowe, 2020*; *Li et al., 2020*).

Immunoglobulin G (IgG) N-linked glycosylation mediates critical functions modulating antiviral immunity during viral infection. Changes in the conserved N-linked glycan Asn297 in the Fc region of IgG, typically by fucosylation, galactosylation, or sialylation, can alter antibody effector function. A reduction in core fucosylation decreases IgG binding to NK cell FcγRIIIa promotes antibody-dependent cellular cytotoxicity (ADCC) necessary for clearance of viruses, including SARS-CoV-2, dengue, and HIV-1, whereas sialylation can attenuate immune responses, resulting in immune evasion (*Ash et al., 2022*; *Haslund-Gourley et al., 2024*; *Hou et al., 2021*; *Wang et al., 2017*). Changes in IgG and other protein N-linked glycosylation profiles, therefore, shape virus-host interactions and disease progression.

Importantly, there have not been prior studies specifically examining plasma IgG N-glycome profiles derived from chromatographic peak data in HFRS patients, particularly in relation to seroconversion status. This gap in our knowledge motivated our systematic investigation of both total and virus-specific IgG glycosylation dynamics during acute infection. In the present study, we combined scRNA-seq and flow cytometry to reveal the phenotypes of B cell responses during HTNV infection and explored the transcriptomic features of reactive B cell subsets. A total of 166 HFRS patients, including 65 paired HFRS samples, were employed to profile the IgG-Fc glycosylation pattern and to investigate the potential regulatory role of IgG-Fc glycosylation in HFRS pathogenesis.

## Results

### B cell compositional characteristics in HFRS patients

To characterize the humoral immune profiles in HFRS patients, we enrolled 166 suspected HTNV-infected patients who were admitted to Baoji Central Hospital in Shaanxi Province, China, between October 2019 and January 2022. Among them, 65 met the inclusion criteria and were included in the study (*Figure 1*). We identified a total of eight cell subpopulations, including subsets of CD4 T cells, CD8 T cells, CD14 monocytes, NK cells, B cells, platelets, endothelial progenitor cells, and red blood cells (*Figure 2—figure supplement 1A and B*). The proportions of these cell types were comparable between acute HFRS patients and healthy controls, with no significant batch effects observed (*Figure 2—figure supplement 1C*). CD4 T cells (31.4%) were the predominant subset

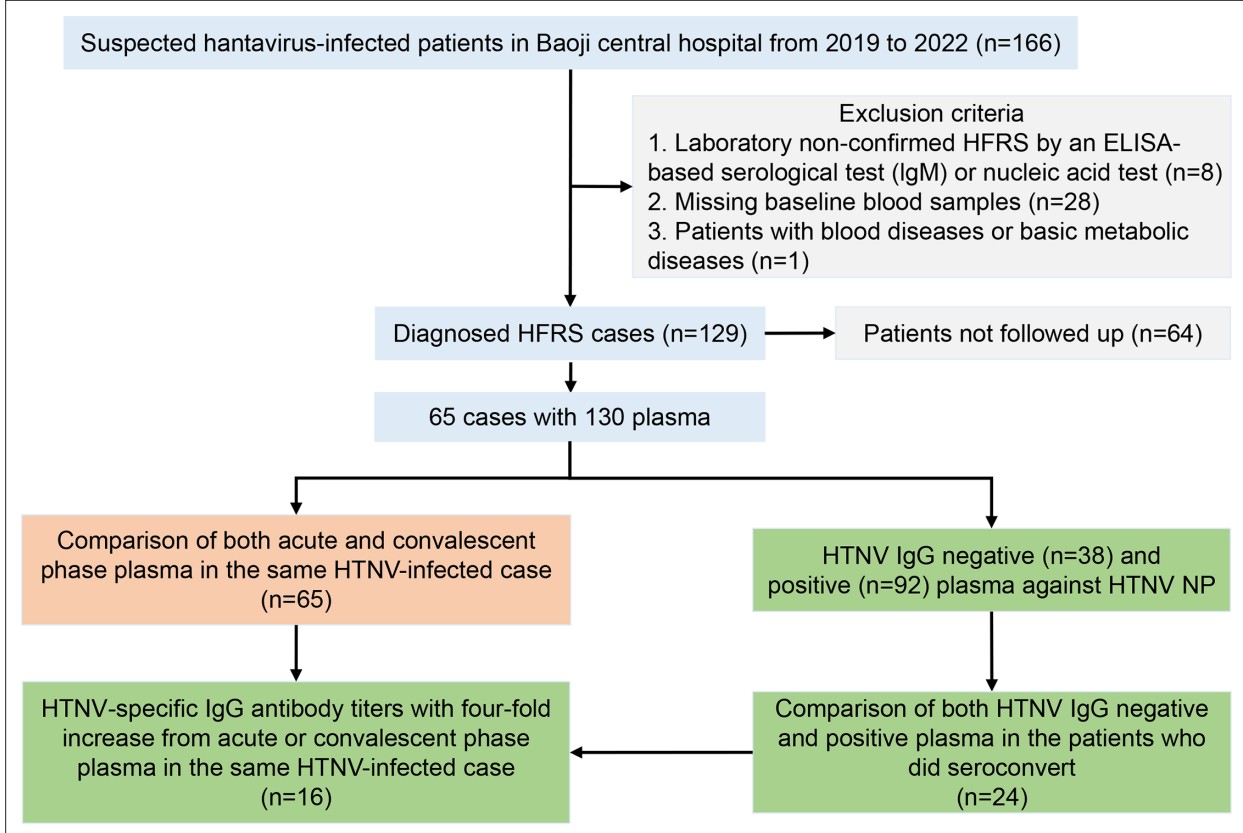

**Figure 1.** Flowchart of Haemorrhagic fever with renal syndrome (HFRS) patient recruitment in Baoji Central Hospital in Shaanxi province from November 2019 to January 2022.

prior to HTNV infection, whereas CD8 T cells (44%) became the dominant population post-infection. Notably, the proportions of NK cells (15.9–5.6%, p<0.001) and platelets (3.1–0.6%, p<0.01) significantly decreased, whereas the level of CD14 monocytes (15.9–32.3%, p<0.05) nearly doubled. The overall proportions of B cells, however, did not exhibit significant changes (*Figure 2—figure supplement 1D and E*).

To precisely elucidate the role of B cell subsets in anti-HTNV infection, we classified B cells into eight subsets, including antibody-secreting memory B cells (ASM), double-negative B cells (DN), intermediate memory B cells (IM), marginal zone-like B cells (MZB), naive B cells (naive B), plasmablasts (PB), quiescent resting memory B cells (RM), and exhausted tissue-like memory B cells (TLM) (*Figure 2A and B*, *Figure 2—figure supplement 1F*). We discovered that significant expansion of ASM, PB, and RM cell populations in acute HFRS patients compared to healthy controls was accompanied by contraction of DN, IM, and naive B cell compartments post-infection (*Figure 2C*). In healthy controls, naive B cells (42.87%), DN B cells (26.65%), and TLM B cells (16.13%) constituted the majority of the B cell population. However, ASM (30.96%), PB cells (11.12%), and RM B cells (2.88%) became the dominant subsets after HTNV infection (*Figure 2D* and *Supplementary file 1*).

To validate the accuracy of our single-cell subpopulation analysis, we employed flow cytometry to quantify the B cell subsets in acute and convalescent HFRS patients (*Figure 2E*). The results demonstrated that ASM (39.48% vs 24.57%, p<0.001) and PB (23.97% vs 2.61%, p<0.001) subsets were significantly amplified in the acute phase compared to the convalescent phase (*Figure 2F*). Conversely, the number of naive B cells (32.78% vs 48.92%, p=0.003) and CSM cells (18.65% vs 27.15%, p=0.008) recovered rapidly in the convalescent stage (*Figure 2F*), consistent with the observations at the transcriptome level. No significant differences were observed for other B cell subtypes between the acute and convalescent stages (*Figure 2—figure supplement 2*). These results demonstrated that several B cell subpopulations were effectively activated following HTNV infection.

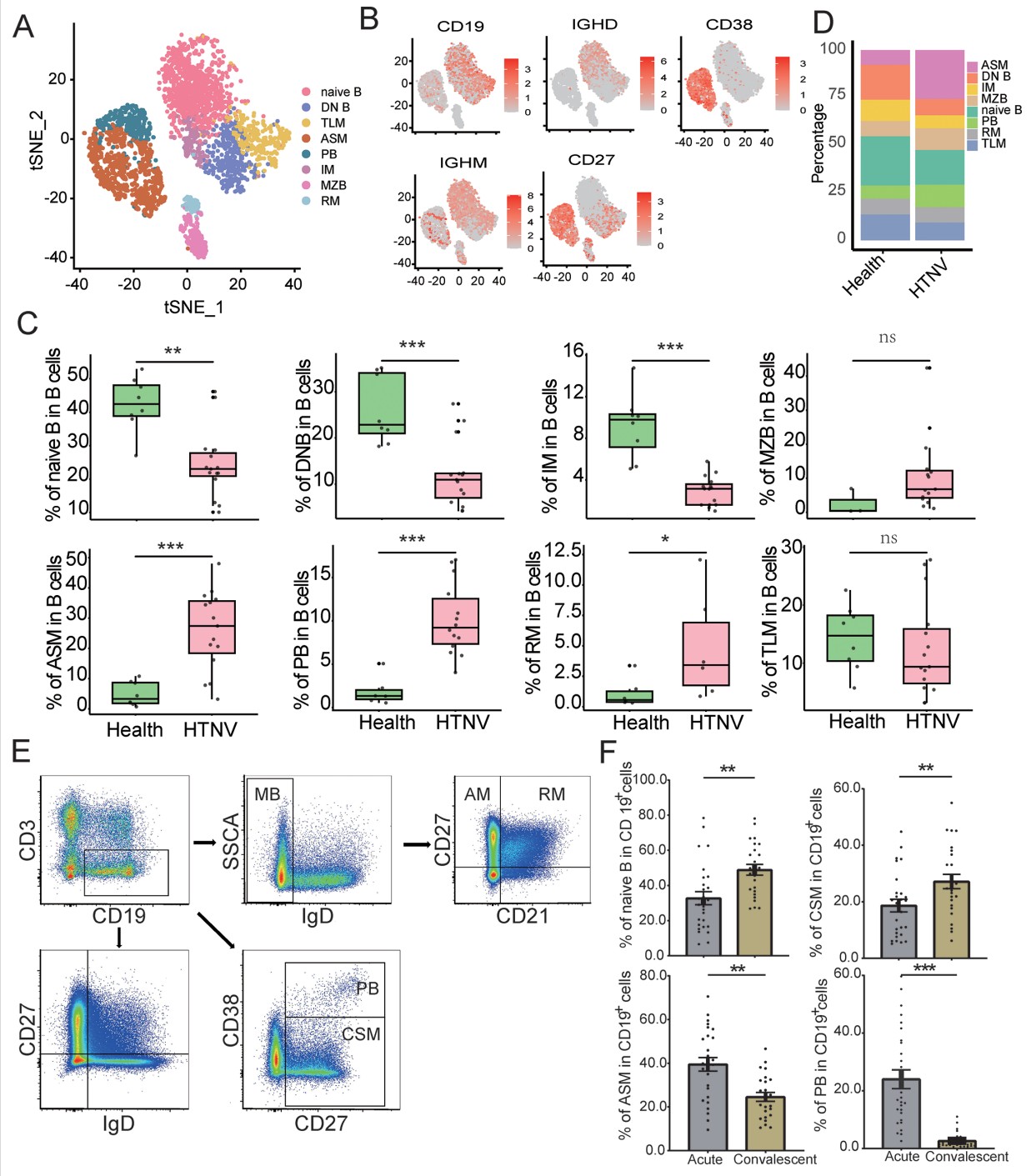

**Figure 2.** Immunophenotypic remodeling in the B cell subsets during acute Haemorrhagic fever with renal syndrome (HFRS). (**A**) t-distributed stochastic neighbor embedding (t-SNE) plot showing antibody-secreting memory B cells (ASM), double-negative (DN) B cells, intermediate memory cells (IM), marginal zone-like cells (MZB), naive B cells, plasmablasts (PB), quiescent resting memory cells (RM), and exhausted tissue-like memory B cells (TLM) of peripheral blood mononuclear cells (PBMCs) identified using an integrated and classification analysis. (**B**) t-SNE projection of canonical markers, including CD19, CD27, CD38, IgD, and IgM. (**C**) Proportions of the eight B cell subsets, colored by the healthy group (green,n=8) and Hantaan virus (HTNV) groups (red, n=15). Boxplot features: minimum box, 25th percentile; center, median; maximum box, 75th percentile.The Wilcoxon signed rank test was used. (**D**) Frequency of the eight B cell subsets in between the healthy group and the HTNV groups. (**E**) FACS gating strategy for the measurement of the B cell subsets: activated memory B cells (CD21- CD27+, AM), RM B cells (CD21+ CD27+), IM B cells (CD21+ CD27-), TLM B cells (CD21- CD27-), naive B cells (CD27- IgD+), MZB B cells (CD27+ IgD+), ASM B cells (CD27+ IgD-), DN B cells (CD27- IgD-), PB (CD38+ CD27+), and class-switched memory B cells (CD38- CD27+, CSM), respectively. (**F**) The proportion of ASM, CSM, naive B, and PB cells in CD19+ B cells in the acute (n=27) and convalescent HFRS

*Figure 2 continued on next page*

*Figure 2 continued*

patients (n=25). The Wilcoxon signed rank test was used. Data are presented as mean ± SEM in panels C and F. *p<0.05,**p<0.01, ***p<0.001, ns, no significane.

The online version of this article includes the following figure supplement(s) for figure 2:

**Figure supplement 1.** Single-cell transcriptomes of peripheral blood mononuclear cells (PBMCs) from patients with Haemorrhagic fever with renal syndrome (HFRS).

**Figure supplement 2.** Dynamic analysis of the B cell subsets in Haemorrhagic fever with renal syndrome (HFRS) patients.

## Dynamic characteristics of the IgG N-glycome in HFRS patients

HFRS patients were further stratified into three age groups (≤44, 45–59, and ≥60 years). IgG glycosylation was analyzed by ultra-performance liquid chromatography. As shown in *Supplementary file 2*, no significant differences in IgG glycosylation were observed between age groups, except for galactosylation (p=0.001). Similarly, no sex-based differences in IgG-Fc glycosylation were noted (*Supplementary file 3*). However, the levels of IgG with bisecting GlcNAc (15.72% vs 14.51%, p<0.001), galactosylated IgG (74.26% vs 71.15%, p<0.001), and sialylated IgG (22.44% vs 21.62%, p<0.001) were significantly higher in the acute phase compared to the convalescent phase. Conversely, fucosylated IgG levels (94.29% vs 94.89%, p<0.001) were lower in the acute phase (*Supplementary file 4*), indicating that IgG N-glycosylation plays a critical role in the anti-infection process. Multivariate linear regression was employed to mitigate potential confounding by genetic and environmental factors in the glycomics analysis. While no significant associations were observed for most glycan models (fucosylation, $p=0.526$; bisecting GlcNAc, $p=0.069$; and sialylation, $p=0.058$), we discovered sex showed a potentially influential effect on galactosylation ($p=0.001$) (*Supplementary files 5–8*). These results suggest that while most glycan features appear unaffected by the examined covariates, galactosylation may be subject to sex-specific biological regulation.

Among the 130 paired blood samples from 65 patients, 38 (29.2%) were negative for HTNV nucleocapsid protein-specific IgG antibodies, and 92 (70.8%) were seropositive. For the 24 patients with both seronegative and seropositive samples, levels of bisecting GlcNAc (16.16% vs 14.54%, $p<0.001$), galactosylation (74.12% vs 70.94%, $p=0.006$), and sialylation (22.92% vs 21.80%, $p=0.008$) were higher in the seronegative period, whereas that of fucosylation was lower (94.36% vs 94.74%, $p=0.024$) in this period (*Figure 3A*). Notably, bisecting GlcNAc and sialylated IgG levels decreased as HTNV nucleocapsid protein (NP)-specific antibody titers increased (*Figure 3B*), highlighting a potential link between antibody levels and IgG-Fc glycosylation.

## The IgG N-glycome in HFRS patients with seroconversion

A fourfold or greater increase in HTNV NP-specific antibody titers usually indicates a protective humoral immune response during the acute phase (*Onwuchekwa et al., 2023*). We observed a significant increase in the Fc fucosylation (94.18% vs 94.48%, $p=0.044$) and a decrease in bisecting GlcNAc (16.41% vs 14.71%, $p=0.001$) post the fourfold antibody titer increase compared to the baseline. However, no significant differences were observed in galactosylation and sialylation (*Figure 3C*), suggesting that fucosylation and bisecting GlcNAc may play key roles in anti-HTNV infection and recovery.

## The potential source of the IgG N-glycome during HTNV infection

Positive correlations were observed between the ASM subsets and both galactosylation ($p=0.017$, $r_s = 0.418$) and sialylation ($p=0.008$, $r_s = 0.458$) in the antibody Fc region, as well as between the PB subsets and sialylation ($p=0.036$, $r_s = 0.372$) (*Figure 4A–C*). Therefore, we speculated that galactosylated antibodies may primarily originate from the ASM subsets, while sialylated antibodies may derive from the ASM and PB subsets.

To investigate the process of glycosylated antibody production secreted by the ASM and PB subsets, we identified differentially expressed genes (DEGs) among the eight B cell subpopulations. Notably, the most DEGs were observed in the ASM (470), naive (312), and TLM (333) B subpopulations, respectively (*Figure 4—figure supplement 1A* and *Supplementary file 9*). Specifically, among all the upregulated genes, the expressions of glycosylation-related, mitochondrial respiratory-related, and peptide chain elongation-related genes among the ASM, PB, and RM populations were elevated

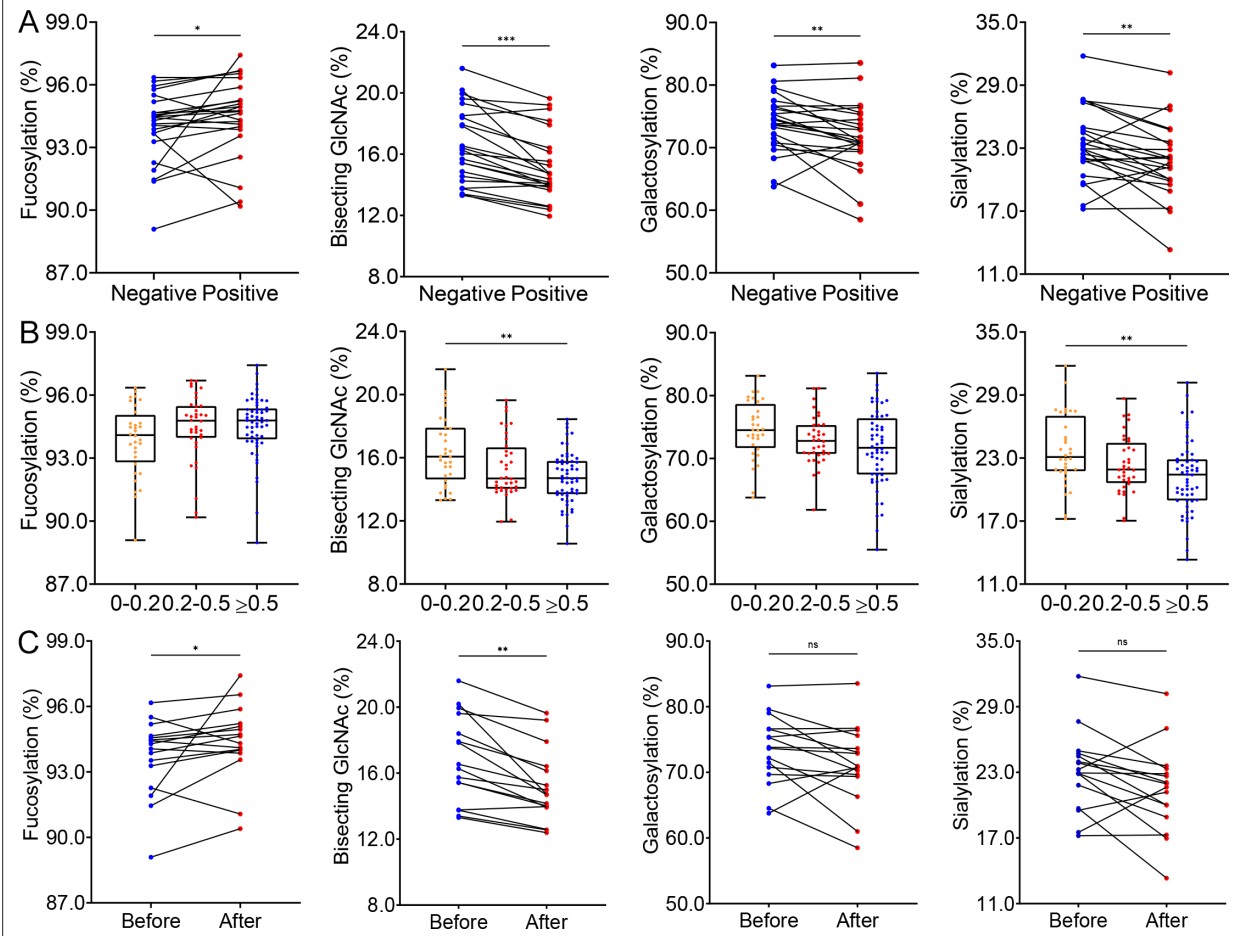

**Figure 3.** Glycosylation modification of antibodies associated with Hantaan virus (HTNV) infection. (**A**) Changes of different glycosylation types in both HTNV NP-specific IgG-negative and positive plasma from 24 Haemorrhagic fever with renal syndrome (HFRS) patients. (**B**) Differential IgG glycosylation patterns across antibody titer levels quantified by enzyme-linked immunosorbent assay (ELISA). (**C**) Comparison of different glycosylation levels before and after the fourfold increase in the IgG antibody titers. *$p<0.05$, **$p<0.01$, ***$p<0.001$,ns, no significance . For paired samples, the Wilcoxon signed rank test was used to assess the difference.

after HTNV infection, including *RPN2*, *MT-ND6*, and *RPS26* (**Figure 4—figure supplement 1B** and **Supplementary file 10**). On the contrary, the upregulated genes in the naive, DN, and IM populations were related to anti-inflammatory and immunosuppressive effects, inhibition of the antioxidative function, response to environmental stress, and B cell growth and development, such as *TSC22D3*, *TXNIP*, *DUSP1*, and *KLF6*, whereas the downregulated genes played an important role in the inflammatory response, cell motility, and antigen presentation, including *LTB*, *ACTG1*, and *HLA-DRB5* (**Figure 4— figure supplement 1C** and **Supplementary file 11**).

Unexpectedly, we observed significant changes in the expression of glycosylation-related genes (GRGs) in the ASM, PB, and RM subclusters during HTNV infection compared to healthy controls. *RPN1* and *RPN2* were prominently upregulated, which encoded subunits of the oligosaccharyltransferase complex (**Figure 4D**). This implied that the process of glycosylation during HTNV infection was rapidly activated among the ASM and PB subclusters. In the ASM subcluster, the fold change of GRGs from all four glycosylation modification-related genes, aside from *RPN1* and *RPN2*, was similar, suggesting that ASM may undergo complex and diverse glycosylation modification processes.

To enhance our understanding of the transcriptomic characteristics of GRG-high-expressing PB and ASM subclusters, we conducted a comprehensive pathway enrichment analysis and identified significant enrichment of pathways related to glycosylation modifications, including the endoplasmic reticulum protein-containing complex, rough endoplasmic reticulum, and protein N-linked glycosylation via asparagine (**Figure 4E**). Additionally, gene sets associated with glycosylation modifications

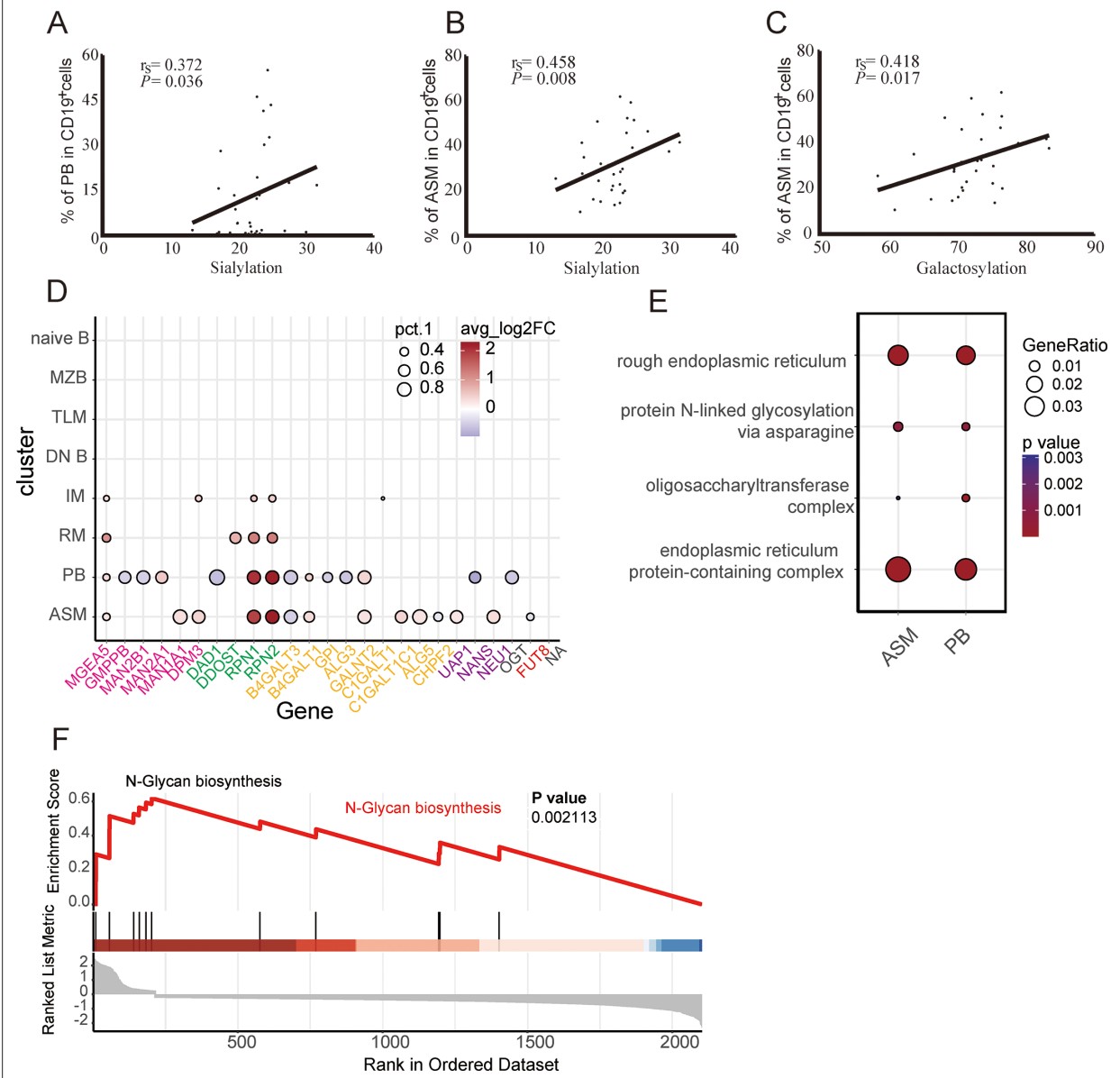

**Figure 4.** Glycosylation modifications of antibodies primarily derived from antibody-secreting and plasmablast subpopulations. (**A**) The correlation between the proportion of plasmablasts (PB) cells and the sialylation level. (**B**) The correlation between the proportion of antibody-secreting memory (ASM) cells and the sialylation level. (**C**) The correlation between the proportion of ASM cells and the galactosylation level. (**D**) Dot plot shows the expression levels of glycosylation-related genes in the eight B cell groups. The pink color represents mannose-related genes, the green color represents N-glycosylation-related genes, the orange color represents galactosylation-related genes, the purple represents sialylation-related genes, and the red color represents fucosylation-related genes. (**E**) Enriched pathways in the plasmablast and ASM subsets by GO enrichment analysis of the differentially expressed genes (DEGs). (**F**) The GSEA map presents the enrichment score of glycosylation-related genes (GRGs) in N-glycan biosynthesis.

The online version of this article includes the following figure supplement(s) for figure 4:

**Figure supplement 1.** The differentially expressed genes and their functional changes in the B cell subsets post-Hantaan virus (HTNV) infection.

showed positive regulation of these pathways (*p*=0.002, *Figure 4F*). These results indicated that glycosylation modifications in the ASM, PB, and RM subpopulations were dynamically regulated during the acute phase of HFRS, likely to satisfy the amplified anti-HTNV antibody requirement and accelerate viral clearance.

## Discussion

IgG glycosylation is a key regulator of inflammatory responses for infectious diseases, acting as a molecular switch between anti-inflammatory and pro-inflammatory effects upon antigenic challenge. One of the most significant findings of the present study is the elevation in IgG fucosylation and reduction in bisecting GlcNAc levels at the time of a fourfold increase in the HTNV NP-specific antibody titers. Afucosylation is known to enhance antibody-dependent cellular cytotoxicity (ADCC) by increasing the affinity of IgG-Fc for the FcγRIIIa receptor on NK cells, leading to heightened production of inflammatory cytokines, such as IL-1β, IL-6, IFN-γ, and TNF-α (*Larsen et al., 2021*; *Lu et al., 2018*; *Wieczorek et al., 2020*). An increase in bisecting GlcNAc linked to Fc induces enhanced affinity for the FcγIIIa receptor and enhanced ADCC function of the antibodies (*Giron et al., 2020*; *Hou et al., 2021*; *Irvine and Alter, 2020*). However, rather than directly influencing the Fc receptor binding, this modification enhances antibody functionality through afucosylation, which subsequently increases the binding affinity to the FcgRIIIa receptor (*Ackerman et al., 2018*; *Jennewein and Alter, 2017*). Our findings provide evidence that the dynamic alterations in glycosylation patterns, characterized by increased fucosylation and decreased bisecting GlcNAc levels during viral infection, are involved in disease pathogenesis and clinical recovery, suggesting their dual utility as both diagnostic biomarkers and promising therapeutic targets.

We also observed elevated galactosylation levels in the acute phase compared to the convalescent phase, a phenomenon also reported in SARS-CoV-2 infections but not in chronic diseases and cancers (*Hou et al., 2021*; *Liu et al., 2018*). Increased galactosylation may expand the distance between CH2 domains, exposing key amino acids for FcγRIIIa binding and enhancing ADCC functions (*Houde et al., 2010*). Simultaneously, the increase in the ASM subset, which positively correlated with galactosylation, may reflect polyclonal activation of memory B cells driven by pattern recognition receptors and cytokines, leading to the production of galactosylated antibodies. Similarly, sialylation levels were higher in the acute patients, potentially linked to the expansion of the ASM and PB subsets. In this study, the PB subset was significantly elevated in the acute phase and positively correlated with sialylation, suggesting that rapidly proliferating plasmablasts may contribute to the secretion of sialylated antibodies; however, the mechanisms behind this require further investigation.

Antibody-related glycogenes are significantly activated following Hantaan virus infection. We noted that ribophorin I and II (RPN1 and RPN2) were significantly upregulated in the ASM/IM/PB/RM subsets after Hantaan virus infection, which linked the high mannose oligosaccharides with asparagine residues found in the Asn-X-Ser/Thr consensus motif (*Hwang et al., 2025*). We speculate that they continuously attach the synthesized glycan chains to the constant region of antibodies during antibody synthesis. Similarly, fucosyltransferase 8 (FUT8) in the ASM subset, catalyzing the alpha1-2, alpha1-3, and alpha1-4 fucose addition (*Wang and Ravetch, 2019*; *Yang et al., 2015*), was downregulated in the mRNA translation, and the levels of fucosylated antibodies were naturally lower in the acute HFRS patients. Meanwhile, the beta-1,4-galactosyltransferase (beta4GalT) gene expression was significantly elevated in the ASM subpopulation during the acute phase, which also correlated with increased levels of galactosylated antibodies in serum (*Wang and Ravetch, 2019*). However, we did not observe significant upward changes in sialyltransferase mRNA expression in the acute HFRS patients, similar to the findings from severe COVID-19 cohorts (*Haslund-Gourley et al., 2024*). The neuraminidase 1 (NEU1) gene is strikingly upregulated and may potentially explain the decreased sialylation on the secreted HTNV-specific IgG antibodies during convalescence. Overall, the glycosylation of immunoglobulin G is regulated by a large network of B cell glycogenes during HTNV infection.

Despite establishing a well-characterized patient cohort and performing systematic IgG glycosylation profiling based on HTNV NP antibody status, this study has several noteworthy limitations. Most notably, while preliminary comparisons suggested similar patterns between virus-specific and total IgG N-glycome, our total plasma IgG analysis may have introduced confounding factors in the observed associations. This methodological constraint could potentially affect the interpretation of certain disease-specific glycosylation signatures.

Our findings demonstrate that the IgG-Fc glycosylation plays a pivotal role in HTNV infection and rapid recovery. Fucosylated oligosaccharides and the absence of bisecting GlcNAc are associated with favorable clinical outcomes. Importantly, specific B cell subsets, particularly ASMs and PBs, upregulated the secretion of galactosylated and sialylated antibodies, respectively. These findings would

deepen our understanding of antibody glycosylation in antivirus immunity and provide a foundation for future studies aimed at optimizing glycoengineered therapeutic antibodies.

## Materials and methods

### Study participants

Clinical specimens were collected from HFRS patients who were hospitalized in Baoji Central Hospital between October 2019 and January 2022. Patients were categorized into four clinical subtypes (mild, moderate, severe, and critical) based on the diagnostic criteria for HFRS issued by the Ministry of Health (*Ma et al., 2015*). This study was approved by the ethics committee of the Shandong First Medical University & Shandong Academy of Medical Sciences (R201937). Written informed consent was obtained from each participant or their guardians.

The clinical course of HFRS is grouped into acute (febrile, hypotensive, and oliguric stages) and convalescent (diuretic and convalescent stages) phases. The acute phase was defined as within 12 days of illness onset, and the convalescent phase was defined as a period of illness lasting 13 days or longer (*Tang et al., 2019*; *Zhang et al., 2022*). The earliest sample was selected if there were multiple blood samples available in the acute phase and the last available sample before discharge was selected if there were multiple blood samples in the convalescent phase.

### Single-cell RNA sequencing and public database

Flow cytometry antibody staining was conducted as previously described (*Chakraborty et al., 2022*). Dead cells and red blood cells were removed using the BD FACSAria III cell sorter, and peripheral blood mononuclear cells (PBMCs) were collected for single-cell sequencing. The library construction was conducted as previously described (*Jin et al., 2024*).

We obtained single-cell transcriptome sequencing data from the Gene Expression Omnibus (GEO) database (GSE161354), containing six HFRS patients and two healthy volunteers, as well as healthy control samples from the Genome Sequence Archive (GSA) database (HRA000203), matched with patient demographic information.

### Single-cell data analysis

The FastQC software was used to evaluate the data obtained to ensure the quality of the raw sequencing data. The raw data were mapped to the human reference genome (GRCh38, https://cf.10xgenomics.com/supp/cell-exp/refdata-gex-GRCh38-2020-A.tar.gz) using Cell Ranger. The Seurat R package (Version 4.4.0) was utilized for the merging and clustering of single-cell data. The Harmony R package and the anchor module of Seurat were used to remove batch effects between samples and groups for cell clustering. t-distributed Stochastic Neighbor Embedding (t-SNE) and Uniform Manifold Approximation and Projection (UMAP) were employed for dimensionality reduction and visualization of individual cells.

### Cell type annotation

UMAP and t-SNE were used to reduce the dimension of all cells and to cluster them in a two-dimensional space based on shared features. The Single R (Version 2.2.0) package was used to infer independently the cell origin of each single cell based on a reference transcriptome dataset of pure cell types, for unbiased cell type identification from single-cell RNA sequencing data. Subsequently, specific highly expressed genes were combined with those derived from a review of the literature for manual annotation to ultimately determine cell types. Classic biomarkers for specific cell types were used to identify cells in different clusters.

### Differential gene identification and functional analysis

The `FindMarkers()` function in the Seurat package was used to identify DEGs between different cell groups, with criteria of |log2FC|>0.25 and *p*-value <0.05, pct.1>0.25. The ClusterProfiler R package was utilized for GO, KEGG, and GSEA enrichment analyses, and the ggplot2 and enrichplot R packages were used for visualization.

## Measurement of HTNV-specific antibodies

Levels of serum IgM and IgG antibodies to HTNV NP were assessed through the enzyme-linked immunosorbent assay (ELISA) kit (Wantai BioPharm, Beijing, China) in accordance with the manufacturer's instructions.

## Analysis of IgG glycans

The process of isolating, labelling, purifying, and analyzing plasma IgG was performed using previously established methods (*Hou et al., 2019*; *Liu et al., 2023*). Briefly, the diluted plasma samples were transferred onto a 96-well protein G monolithic plate (BIA Separations, Slovenia) for the isolation of IgG. The isolated IgG was eluted with 1 mL of 0.1 M formic acid and was immediately neutralized with 170 µL of 1 M ammonium bicarbonate.

The released N-glycans were labelled with 2-aminobenzamide (2-AB) and were then purified from a mixture of 100% acetonitrile and ultrapure water in a 1:1 ratio (v/v). This was then analyzed by hydrophilic interaction liquid chromatography using ultra-performance liquid chromatography (HILIC-UPLC; Walters Corporation, Milford, MA) (*Hou et al., 2019*). As previously reported, the chromatograms were separated into 24 IgG glycan peaks (GPs) (*Menni et al., 2018*).

Our study incorporated both biological and technical replicates to ensure a robust glycomic profiling analysis. Specifically, we analyzed paired acute/convalescent-phase samples from 65 confirmed HFRS patients to assess inter-individual biological variability, while technical reproducibility was validated through comparison with standard chromatographic peak plots (*Vučković et al., 2016*). This dual-replicate strategy enabled a comprehensive evaluation of both biological heterogeneity and assay precision, and the coefficient of variation for samples were below 16%.

## Virus-specific B cell detection

Virus-specific B cells against HTNV were performed as previously described (*Liechti and Roederer, 2019*; *Song et al., 2018*). All antibodies used for flow cytometry were purchased from BioLegend or BD Biosciences. Antibody combination was as follows: CD3-FITC (cat, 300406), CD19-PerCP-Cy5.5 (cat, 561295), CD21-APC (cat, 559867), CD27-DAPI (cat, 562513), CD38-PE (cat, 555460), IgD-Amcyan (cat, 563034), IgG-PE-CY7 (cat, 561298), and IgM-BV605 (cat, 562977).

## Statistical analysis

Categorical variables were compared using chi-square tests. Normality of numeric variables was assessed with the Kolmogorov-Smirnov test; non-normally distributed continuous variables were analyzed using non-parametric tests. The Benjamini - Hochberg (BH) method was used to adjust the raw p-values from DEG analysis, controlling the false discovery rate (FDR). Statistical analyses were performed with R software version 4.1.1 (R Core Team, New Zealand) and GraphPad Prism 8 (GraphPad Software, San Diego, CA), with a significance level of $\alpha=0.05$ (two-tailed).

# Additional information

## Funding

| Funder | Grant reference number | Author |
| --- | --- | --- |
| The Natural Science Foundation of Shandong Province | ZR2020QH133 | Chuansong Quan |
| The Natural Science Foundation of Shandong Province | ZR2022MH082 | Haifeng Hou |
| The Natural Science Foundation of Shandong Province | ZR2024MH235 | Weijia Xing |

| Funder | Grant reference number | Author |
|---|---|---|
| The Natural Science Foundation of Shandong Province | ZR201911090028 | Yaoni Li |
| Shananxi Provincial Natural Science Basic Research Program of Shaanxi Province | 2023-JC-YB-741 | Yaoni Li |

The funders had no role in study design, data collection and interpretation, or the decision to submit the work for publication.

## Author contributions

Chuansong Quan, Data curation, Funding acquisition, Writing – original draft, Writing – review and editing; Lu Wang, Data curation, Software, Formal analysis, Investigation, Writing – original draft; Jiming Gao, Data curation, Software, Methodology; Yaoni Li, Resources, Funding acquisition, Investigation; Xiaoyu Xu, Software, Formal analysis; Houqiang Li, Zixuan Gao, Wenxu Ruan, Hongzhi Liu, Investigation; Qian Li, Data curation, Investigation; Weijia Xing, Funding acquisition, Investigation; Liqiong Zhao, Resources, Investigation; Michael J Carr, Writing – review and editing; Weifeng Shi, Conceptualization, Supervision, Writing – review and editing; Haifeng Hou, Conceptualization, Supervision, Funding acquisition, Writing – review and editing

## Author ORCIDs

Chuansong Quan http://orcid.org/0000-0002-9610-2567
Lu Wang http://orcid.org/0009-0009-5366-9748
Jiming Gao http://orcid.org/0000-0003-4347-4049
Yaoni Li http://orcid.org/0000-0002-6326-6004
Zixuan Gao http://orcid.org/0009-0007-9435-3585
Wenxu Ruan http://orcid.org/0009-0003-3488-4369
Qian Li http://orcid.org/0000-0002-7808-289X
Weijia Xing http://orcid.org/0000-0003-0252-6340
Liqiong Zhao http://orcid.org/0009-0009-6139-2961
Weifeng Shi https://orcid.org/0000-0002-8717-2942
Haifeng Hou https://orcid.org/0000-0002-1131-1619

Reviewer #1 (Public review): https://doi.org/10.7554/eLife.106989.4.sa1
Reviewer #2 (Public review): https://doi.org/10.7554/eLife.106989.4.sa2
Author response https://doi.org/10.7554/eLife.106989.4.sa3

# Additional files

## Supplementary files

Supplementary file 1. The proportion of different B cell subpopulations between healthy and acute Haemorrhagic fever with renal syndrome (HFRS) groups.

Supplementary file 2. Difference in main IgG glycome features of Haemorrhagic fever with renal syndrome (HFRS) patients between age groups.

Supplementary file 3. Difference in main IgG glycome features of Haemorrhagic fever with renal syndrome (HFRS) patients between sex groups.

Supplementary file 4. Relative abundance (%) of the main IgG-Fc glycome features in Haemorrhagic fever with renal syndrome (HFRS) patients.

Supplementary file 5. Results of multiple linear regression analysis for fucosylation. APTT, activated partial thromboplastin time; MCHC, mean corpuscular hemoglobin concentration; NEUT, neutrophil; PLT, platelet; WBC, white blood cell; CRE, creatinine; Cys-C, cystatin C.

Supplementary file 6. Results of multiple linear regression analysis for bisecting GlcNAc. APTT, activated partial thromboplastin time; MCHC, mean corpuscular hemoglobin concentration; NEUT, neutrophil; PLT, platelet; WBC, white blood cell; CRE, creatinine; Cys-C, cystatin C.

Supplementary file 7. Results of multiple linear regression analysis for galactosylation. APTT,

activated partial thromboplastin time; MCHC, mean corpuscular hemoglobin concentration; NEUT, neutrophil; PLT, platelet; WBC, white blood cell; CRE, creatinine; Cys-C, cystatin C.

Supplementary file 8. Results of multiple linear regression analysis for sialylation. APTT, activated partial thromboplastin time; MCHC, mean corpuscular hemoglobin concentration; NEUT, neutrophil; PLT, platelet; WBC, white blood cell; CRE, creatinine; Cys-C, cystatin C.

Supplementary file 9. Differentially expressed genes among different B cell subgroups in healthy and acute Haemorrhagic fever with renal syndrome (HFRS) groups.

Supplementary file 10. Differential expression genes in the antibody-secreting memory B cells (ASM), plasmablasts (PB), and resting memory (RM) B cell subpopulations.

Supplementary file 11. Differential expression genes in the DN, IM, and naive B cell subpopulations.

MDAR checklist

### Data availability

The raw RNAseq data used for this study is available at the BioProject database (the accession number: PRJNA1238574).

The following dataset was generated:

| Author(s) | Year | Dataset title | Dataset URL | Database and Identifier |
|---|---|---|---|---|
| Quan C, Wang L, Gao J, Li Y, Xu X, Li H, Gao Z, Ruan W, Liu H, Li Q, Xing W, Zhao L, Carr MJ, Shi W, Hou H | 2025 | Glycosylated IgG antibodies accelerated the recovery of haemorrhagic fever with renal syndrome patients | https://www.ncbi.nlm. nih.gov/bioproject/ PRJNA1238574 | NCBI BioProject, PRJNA1238574 |

The following previously published datasets were used:

| Author(s) | Year | Dataset title | Dataset URL | Database and Identifier |
|---|---|---|---|---|
| Zhang Y, Tang K | 2021 | A single-cell atlas of the peripheral immune response in patients with HFRS | https://www.ncbi. nlm.nih.gov/geo/ query/acc.cgi?acc= GSE161354 | NCBI Gene Expression Omnibus, GSE161354 |
| Su W | 2020 | A human immune cell landscape in aging and COVID-19 | https://ngdc.cncb. ac.cn/gsa-human/ browse/HRA000203 | Genome Sequence Archive, HRA000203 |

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
