## [Editor Report · eLife Assessment]

The authors investigated the potential role of IgG N-glycosylation in Haemorrhagic Fever with Renal Syndrome (HFRS), which may offer significant insights for understanding molecular mechanisms and for the development of therapeutic strategies for this infectious disease. The findings are **valuable** to the field and the strength of evidence to support the findings is **solid**.

---

## [Referee Report · Reviewer #1 (Public review)]

The authors investigated the potential role of IgG N-glycosylation in Haemorrhagic Fever with Renal Syndrome (HFRS), which may offer significant insights for understanding molecular mechanisms and for the development of therapeutic strategies for this infectious disease.

---

## [Referee Report · Reviewer #2 (Public review)]

This work sought to explore antibody responses in the context of hemorrhagic fever with renal syndrome (HFRS) - a severe disease caused by Hantaan virus infection. Little is known about the characteristics or functional relevance of IgG Fc glycosylation in HFRS. To address this gap, the authors analyzed samples from 65 patients with HFRS spanning the acute and convalescent phases of disease via IgG Fc glycan analysis, scRNAseq, and flow cytometry. The authors observed changes in Fc glycosylation (increased fucosylation and decreased bisection) coinciding with a 4-fold or greater increased in Haantan virus-specific antibody titer. The study also includes exploratory analyses linking IgG glycan profiles to glycosylation-related gene expression in distinct B cell subsets, using single-cell transcriptomics. Overall, this is an interesting study that combines serological profiling with transcriptomic data to shed light on humoral immune responses in an underexplored infectious disease. The integration of Fc glycosylation data with single-cell transcriptomic data is a strength.

---

## [Author Response]

The following is the authors’ response to the previous reviews

**Reviewers 1:**
Summary:The authors investigated the potential role of IgG N-glycosylation in Haemorrhagic Fever with Renal Syndrome (HFRS), which may offer significant insights for understanding molecular mechanisms and for the development of therapeutic strategies for this infectious disease.While the majority of the issues have been addressed, a few minor points still remain unresolved. Quality control should be conducted prior to the analysis of clinical samples. However, the coefficient of variation (CV) value was not provided for the paired acute and convalescent-phase samples from 65 confirmed HFRS patients, which were analyzed to assess inter-individual biological variability. It is important to note that biological replication should be evaluated using general samples, such as standard serum.

We thank the reviewer for this insightful and critical comment regarding the quality control of our analytical data and the assessment of biological variability. We agree that this is essential for validating the reliability of our findings. We have now provided the requested CV data and clarified this point in the revised manuscript as detailed below.

"This dual-replicate strategy enabled a comprehensive evaluation of both biological heterogeneity and assay precision, and the coefficient of variation for samples were below 16%." Please see the Materials and Methods (Page 16, lines 360-362, and Author response table 1).

**Author response table 1. sa3table1:** Comparative analysis of serum biomarker concentrations in acute and convalescent phase cohorts.

Biomarker	Number	Mean	Standard deviation	Coefficient of variation	Phase
Fuc1	65	94.03	1.51	1.61	AcuteConvalescent
Bis1	65	15.91	2.00	12.57	
Gal1	65	74.16	4.57	6.17	
Sial	65	22.92	3.15	13.73	
Fuc2	65	94.59	1.60	1.69	
Bis2	65	14.82	1.83	12.38	
Gal2	65	71.78	5.58	7.78	
Sia2	65	21.53	3.39	15.74	

**Reviewers 2:**
This work sought to explore antibody responses in the context of hemorrhagic fever with renal syndrome (HFRS) - a severe disease caused by Hantaan virus infection. Little is known about the characteristics or functional relevance of IgG Fc glycosylation in HFRS. To address this gap, the authors analyzed samples from 65 patients with HFRS spanning the acute and convalescent phases of disease via IgG Fc glycan analysis, scRNAseq, and flow cytometry. The authors observed changes in Fc glycosylation (increased fucosylation and decreased bisection) coinciding with a 4-fold or greater increased in Haantan virus-specific antibody titer. The study also includes exploratory analyses linking IgG glycan profiles to glycosylation-related gene expression in distinct B cell subsets, using single-cell transcriptomics. Overall, this is an interesting study that combines serological profiling with transcriptomic data to shed light on humoral immune responses in an underexplored infectious disease. The integration of Fc glycosylation data with single-cell transcriptomic data is a strength.The authors have addressed the major concerns from the initial review. However, one point to emphasize is that the data are correlative. While the associations between Fc glycosylation changes and recovery are intriguing, the evidence does not establish causation. This is not a weakness, as correlative studies can still be highly valuable and informative. However, the manuscript would be strengthened by making this distinction clear, particularly in the title.The verb "accelerated" in the title implies that the glycosylation state of IgG was a direct driver of recovery, rather than something that correlated with recovery. Thus, a more neutral word/phrase would be ideal.

We sincerely thank the reviewer for this insightful suggestion. We agree that the use of "accelerated" might overstate the potential role of IgG glycosylation, which has not been clearly clarified by our current findings. As reported in results (particularly in Figure 2), partial glycosylation exhibits statistically significant variations between seropositive and seronegative statuses, before and after seroconversion, and across different HTNV- NP specific antibody titers. Therefore, we have replaced "accelerated" with "contribute to" in the Title: "Glycosylated IgG antibodies contribute to the recovery of haemorrhagic fever with renal syndrome patients".